# Development and Characterization of Quercetin-Loaded Delivery Systems for Increasing Its Bioavailability in Cervical Cancer Cells

**DOI:** 10.3390/pharmaceutics15030936

**Published:** 2023-03-14

**Authors:** Miguel Ferreira, Diana Gomes, Miguel Neto, Luís A. Passarinha, Diana Costa, Ângela Sousa

**Affiliations:** 1CICS-UBI—Health Science Research Centre, University of Beira Interior, Av. Infante D. Henrique, 6200-506 Covilhã, Portugal; 2Associate Laboratory i4HB—Institute for Health and Bioeconomy, NOVA School of Science and Technology, Universidade NOVA, 2819-516 Caparica, Portugal; 3UCIBIO—Applied Molecular Biosciences Unit, Departament of Chemistry, NOVA School of Science and Technology, Universidade NOVA de Lisboa, 2829-516 Caparica, Portugal; 4Laboratório de Fármaco-Toxicologia-UBIMedical, Universidade da Beira Interior, 6200-284 Covilhã, Portugal

**Keywords:** chitosan, cyclodextrin, delivery systems, inclusion complex, quercetin

## Abstract

Quercetin is a natural flavonoid with high anticancer activity, especially for related-HPV cancers such as cervical cancer. However, quercetin exhibits a reduced aqueous solubility and stability, resulting in a low bioavailability that limits its therapeutic use. In this study, chitosan/sulfonyl-ether-β-cyclodextrin (SBE-β-CD)-conjugated delivery systems have been explored in order to increase quercetin loading capacity, carriage, solubility and consequently bioavailability in cervical cancer cells. SBE-β-CD/quercetin inclusion complexes were tested as well as chitosan/SBE-β-CD/quercetin-conjugated delivery systems, using two types of chitosan differing in molecular weight. Regarding characterization studies, HMW chitosan/SBE-β-CD/quercetin formulations have demonstrated the best results, which are obtaining nanoparticle sizes of 272.07 ± 2.87 nm, a polydispersity index (PdI) of 0.287 ± 0.011, a zeta potential of +38.0 ± 1.34 mV and an encapsulation efficiency of approximately 99.9%. In vitro release studies were also performed for 5 kDa chitosan formulations, indicating a quercetin release of 9.6% and 57.53% at pH 7.4 and 5.8, respectively. IC_50_ values on HeLa cells indicated an increased cytotoxic effect with HMW chitosan/SBE-β-CD/quercetin delivery systems (43.55 μM), suggesting a remarkable improvement of quercetin bioavailability.

## 1. Introduction

Flavonoids are a family of natural phenolic compounds widely present in plants, fruits and vegetables; they are characterized by possessing a C6-C3-C6 carbon skeleton base. These natural compounds have emerged as important anticancer agents as it has already been reported that they possess several properties in this way, namely apoptosis induction, the reduction of oxidative stress, the inhibition of angiogenesis and an increased DNA repair process, among others [1,2,3]. Quercetin, the main flavonoid explored in therapeutic applications, has already been studied for different types of cancer, and a higher therapeutic effect for human papillomavirus (HPV)-related cancers due to E6 oncoprotein inhibition capacity has been reported [4,5,6,7]. HPV is a virus with a high capacity to induce proliferative lesions in the skin and internal mucosa, mainly associated with the development of cervical cancer, which are responsible for 79 to 100% of cases [8].

Cervical cancer is the fourth most common type of cancer in women worldwide, leading to over 340,000 deaths in 2020 [9]. Despite existing preventive vaccines against the major types of HPV, they are not widely available and administered in less developed countries due to weaknesses in health systems and anti-vaccination movements. In this case, cervical cancer appears as the second leading cause of cancer and death in women [5,10].

The oncogenic role of HPV is highly associated with the activity of two viral oncoproteins, E6 and E7 [8]. The main function of these oncoproteins is the ability to inhibit p53 and pRb tumor suppressor pathways, respectively [8]. Therefore, the absence of these two tumor suppressor proteins results in the immortalization and cellular transformation of infected cells [11,12,13,14]. Currently, cervical cancer treatment includes surgery, radiation and chemotherapy, which have demonstrated a very limited success rate, especially for the most severe cases and secondary effects in patients, as well as the high cost that limits its utilization in less developed countries [6,15]. Thus, new approaches are being studied in order to develop an effective, specific and cheaper therapy for cervical cancer [16]. These therapies may involve the use of DNA- or RNA-based gene therapies, which have shown great potential, but are not approved yet and can be expensive [6]. In this context, other treatments, such as the use of natural compounds, have been studied due to their low cost, low toxicity and targeted and specific action when compared with other molecules.

Considering what was previously reported, quercetin emerges as a therapeutic agent that can interact with E6 oncoprotein, inhibiting its function, and consequently, leading to an increase in p53 protein levels and apoptosis induction [1,5]. Despite already proven anticancer effects, quercetin shows a low aqueous solubility, low stability, low intestinal permeability and easy degradation in an acidic medium, resulting in low bioavailability, which limits its therapeutic uses [17]. For that reason, new approaches are being developed to increase quercetin bioavailability, where the use of delivery systems is one of the most relevant [5].

Inclusion complexes have been widely reported for the encapsulation of hydrophobic drugs, presenting a high encapsulation efficiency as well as improved drug stability and solubility [5,18]. These inclusion complexes-based delivery systems present a structural cone shape with an outer hydrophilic surface and an inner hydrophobic cavity, thereby being able to “imprison” the hydrophobic drug within the inclusion complexes [18,19]. Cyclodextrins are the most common delivery systems type based on inclusion complexes, with β-Cyclodextrin and its derivatives being the most reported [5]. Although β-Cyclodextrins have relevant drug encapsulation and protection capacities, they form particles with high dimensions, normally exceeding 200 nm, limiting its utilization. Therefore, to overcome this restriction, some physical and chemical modifications and/or conjugation with other molecules, such as polymers, are necessary [5,19,20,21,22].

Sulfobutyl-ether-β-Cyclodextrin (SBE-β-CD) is a modified cyclodextrin that presents a negative surface charge, allowing the conjugation with a positively charged polymer, such as chitosan. This combination can result in a substantial decrease in the size of the delivery system, as well as an increase in the drug stability and bioavailability [20,21]. Chitosan is a natural cationic polymer that has been widely used in delivery systems engineering, due to its high biocompatibility and biodegradability properties as well as low toxicity and mucoadhesion [23,24,25]. The use of this polymer as a delivery system stabilizer has been widely reported, and several types of chitosan with different molecular weights (MW) have already been tested [26]. In this way, the present work explored and optimized SBE-β-CD-based inclusion complexes conjugated with different types of chitosan to encapsulate quercetin, aiming at an increase in quercetin loading capacity, carriage, solubility and, consequently, bioavailability in cervical cancer cells.

## 2. Materials and Methods

### 2.1. Materials

Five kDa MW and low MW (LMW), with an MW range between 50 and 190 kDa, chitosan polymers were purchased from Sigma Aldrich Chemicals (St. Louis, MO, USA). High MW (HMW) chitosan, with an MW range between 200 and 500 kDa, was acquired from Heppe Medical (Halle, Germany). Sodium sulfobutyl ether beta-cyclodextrin (SBE-β-CD) was acquired from Acros Organics (Thermo Fisher Scientific, Waltham, MA, USA). Quercetin was kindly given by Professor Ana Paula Duarte (from CICS-UBI, Portugal).

HeLa cells were obtained from PromoCell (Heidelberg, Germany). DMEM/F12 cell culture media was purchased from Gibco (Thermo Fisher Scientific, Waltham, MA, USA). 3-(4,5-dimethylthiazol-2-yl)-2,5-diphenyltetrazolium bromide (MTT) was obtained from Alfa Aesar (Waltham, MA, USA).

All solutions were freshly prepared by using ultra-pure grade water, purified with a Milli-Q system from Millipore (Billerica, MA, USA).

### 2.2. Methods

#### 2.2.1. Preparation of SBE-β-CD/Quercetin Inclusion Complexes

Inclusion complexes made of SBE-β-CD with encapsulated quercetin were formulated by mixing 30 mg of SBE-β-CD and 0.3 mg of quercetin powder in 5 mL of ultrapure water as described by H. Nguyen and F. Goycoolea [20]. The solution was left stirring at 500 rpm for 24 h and 37 °C, protected from light. Phase-solubility studies performed by H. Nguyen and F. Goycoolea were considered in order to guarantee that the maximum amount of quercetin was encapsulated in the hydrophobic cavity of inclusion complexes [20].

After the formulation step, these inclusion complexes were left at stabilizing for 30 min and were centrifuged at 12,000 rpm for 10 min and 4 °C, before the characterization studies were performed [20].

#### 2.2.2. Preparation of Chitosan/SBE-β-CD/Quercetin Delivery Systems

Stock solutions of chitosan polymers (2 mg/mL) were prepared by dissolving the respective chitosan powder (HMW, LMW or 5 kDa) in 2% acetic acid (pH 3.5). The solutions were left stirring overnight at 250 rpm and then filtered with a 450 nm filter and stored at room temperature until use. SBE-β-CD/quercetin solution was prepared as described above. The conjugation and formulation of chitosan/SBE-β-CD/quercetin delivery systems were performed by ionotropic gelation technique [20,21]. Briefly, a range of different concentrations of SBE-β-CD/quercetin between 6 and 1.5 mg/mL were added, drop by drop, into a solution with different types and/or concentrations (between 2 and 0.5 mg/mL) of chitosan [20,21]. The volumes of each solution were kept constant, being 300 µL in the case of chitosan and 100 µL in the case of SBE-β-CD/quercetin solution.

Before the characterization studies, these delivery systems were left stabilizing for 30 min at room temperature and were centrifuged at 10,000 rpm for 20 min and 4 °C.

#### 2.2.3. UV/vis Absorbance Spectrum

After the formulation and centrifugation of each delivery system, the pellets were resuspended in water and UV–vis spectra were acquired using a UV–vis spectrophotometer (Thermo Scientific™ Evolution 220, Waltham, MA, USA) with a range between 300 and 500 cm^−1^. The same analysis was performed for each isolated component in order to understand potential interactions between components present in each formulation.

#### 2.2.4. Physico-Chemical Characterization Studies

Each type of explored delivery system (SBE-β-CD/quercetin inclusion complex, HMW chitosan/SBE-β-CD/quercetin, LMW chitosan/SBE-β-CD/quercetin and 5 kDa chitosan/SBE-β-CD/quercetin) was formulated by using different concentrations of each component. The physico-chemical properties of each experiment, such as the size, polydispersity index (PdI) and zeta potential, were evaluated by Dynamic Light Scattering (DLS) technique using a Zetasizer Nano ZS equipment (Malvern Instruments, Malvern, UK). All the parameters were measured three times from three independent samples (n = 3).

#### 2.2.5. Encapsulation Efficiency

The quercetin encapsulation efficiency for each system was determined using UV–vis spectrophotometer (Thermo Scientific™ Evolution 220, Waltham, MA, USA) at an ultraviolet wavelength of 374 nm, as described by M. Sundararajan and coworkers with slight modifications [22]. Briefly, each type of delivery system was centrifuged as described above and resuspended in methanol to disrupt the delivery system. Then, samples were sonicated for 10 min and centrifuged for 5 min at 8000 rpm and 4 °C. Finally, the supernatant was recovered, and the quercetin content was quantified using a calibration curve from 5 µM to 100 µM of quercetin dissolved in methanol.

The encapsulation efficiency was determined by using Equation (1). All the conditions were performed in three independent assays (n = 3).
(1)$$Encapsulation\;efficiency\;(\%)=\frac{Concentration\;of\;encapsulated\;drug}{Concentration\;of\;total\;drug}\times 100$$

#### 2.2.6. Scanning Electron Microscopy

To evaluate the morphology of each type of quercetin-loaded delivery system, scanning electron microscopy (SEM) was used. Each type of delivery system was centrifuged as already described and the pellet was resuspended in 200 µL of ultra-pure water followed by centrifugation at 9500 rpm for 12 min at 4 °C. This washing step was repeated three times to ensure that all the impurities were removed. After the last centrifugation, the supernatant was removed, and the delivery systems were resuspended in 40 µL of tungsten 2%. Each sample was then diluted 1:20 in ultra-pure water and 10 µL was placed in a roundly shaped coverslip. The samples were left overnight to dry at room temperature.

The next day, samples were sputter coated with gold using an Emitech K550 (London, UK) sputter coater. SEM Hitachi S-2700 (Tokyo, Japan) was used with an acceleration of 20 kV at various magnifications to evaluate the morphology of each type of delivery system.

#### 2.2.7. Fourier Transform Infrared Spectroscopy

The interaction between compounds present in each delivery system type was evaluated using fourier transform infrared spectroscopy (FTIR). To prepare the samples, each type of formulation was centrifuged, and the pellet was resuspended in ultra-pure water. Each formulation was then freeze-dried at −80 °C followed by a lyophilization process of 24 h using a ScanVac Coolsafe freeze dryer (Labogene, Lillerød, DK).

The spectra of each isolated component and each prepared delivery system were then acquired using a Nicolet iD10 FTIR spectrophotometer (Thermo Scientific, Waltham, MA, USA) with an average of 120 scans, a spectral resolution of 32 cm^−1^ and a spectral width ranging from 4000 and 400 cm^−1^.

#### 2.2.8. Release Studies

Release studies were performed in a phosphate buffer saline (PBS):methanol (80:20 *v*/*v*) solution to simulate the cellular physiological conditions. Methanol was used to ensure the sink conditions, allowing quercetin solubility [27]. For this purpose, the best quercetin-loaded delivery system ratio defined by the characterization studies for each type of formulation was centrifuged and resuspended in a PBS: methanol solution. The quercetin-loaded delivery system solution was then pipetted into a dialysis cassette and was placed in 10 mL of PBS: methanol solution that was left stirring at 100 rpm and 37 °C for 96 h.

At defined time intervals of 0, 0.25, 0.5, 1, 2, 4, 8, 24, 48, 72 and 96 h, 1 mL of the receptor phase was removed and replaced with fresh PBS: methanol solution. The amount of quercetin released from delivery systems was then quantified using a UV–vis spectrophotometer (Thermo Scientific™ Evolution 220, Waltham, MA, USA) at a wavelength of 374 nm using a calibration curve from 1 µM to 100 µM of quercetin dissolved in PBS: methanol solution (80:20 *v*/*v*).

These assays were performed at two different pH (7.4 and 5.8) to understand the release in normal physiological conditions and in acidic conditions, representative of a tumor environment.

#### 2.2.9. Cell Culture

HeLa cells (cervical cancer cells HPV18 positive) were cultured in Dulbecco’s Modified Eagle’s Medium/Ham’s F-12 nutrient mixture (DMEM-F12), supplemented with 10% (*v*/*v*) fetal bovine serum and a mixture of penicillin (100 mg/mL) and streptomycin (100 mg/mL). Cells were grown in 25 cm^3^ T-flasks at 37 °C and in a 5% CO_2_ humidified atmosphere until 80% confluence was obtained.

#### 2.2.10. Cell Internalization

FITC-labelled chitosan was produced as described by Yuqing Ge and coworkers with slight modifications [28]. Briefly, 2 µL of FITC 100 mg/mL dissolved in DMSO was added to 20 mL of each chitosan type (1% *w*/*v* in 2% acetic acid pH 3.5). The reaction was kept for 3 h in dark conditions. After that, each chitosan type solution was washed with distilled water and centrifuged for 30 min at 12.000 rpm at 4 °C until no fluorescence was detected in the supernatant. The delivery systems assemblance was performed as previously described in Section 2.2.2.

HeLa cells were seeded in an 8-well µ-slide (Ibidi, Martinsried, Munich, Germany) at a concentration of 15.000 cells/well for 24 h. Therefore, media was discarded and replaced by free media (0% FBS) for an overnight incubation. At the next day, FITC labelled chitosan/SBE-β-CD/quercetin systems were resuspended in free media and applied to the cells. After 2 h of transfection, cells were visualized using LSM 710 Confocal Laser Scanning Microscope (Carl Zeiss, Oberkochen, Germany) under 63× magnification. To better understand cells’ locations, nuclei were stained with DAPI.

#### 2.2.11. Cell Viability Assays

Cell viability assays were performed by the MTT method, allowing us to evaluate the metabolic activity of cells by the formation of formazan crystals. Briefly, 5 × 10^3^ HeLa cells were seeded in 96-well plates for 24 h. Afterwards, the medium was discarded and 100 µL of fresh medium containing concentrations from 0.1 µM to 150 µM of quercetin-loaded delivery systems was added to each well. A stock solution of 25 mM free quercetin dissolved in 100% DMSO was prepared and diluted in culture medium for the same tested concentrations (assuring DMSO % lower than 1%) as positive control. After 48 h of incubation, the medium was removed, and cells were washed with 100 µL of PBS to remove all the impurities. Then, MTT solution (0.5 mg/mL) was prepared by dissolving MTT powder in serum-free medium and 100 µL was added to each well, followed by incubation for 4 h at 37 °C.

After the incubation time, the medium was removed and 100 µL of dimethyl sulfoxide (DMSO) was added to dissolve formazan crystals. The redox activity was quantified through the absorbance measured at 570 nm, using the microplate reader Bio-Rad xMark spectrophotometer (Bio-Rad, EUA). Cell viability values were presented as percentages relative to the absorbance observed in non-treated cells. All conditions were tested three independent times with four replicates (n = 3).

#### 2.2.12. Statistical Analysis

Each experimental condition was performed in three independent cellular preparations and was expressed as mean ± standard deviation. Values of *p*-value ≤ 0.05 were considered significant. The dose–response curves were built using a sigmoidal dose–response (variable slope) curve fit with a 95% confidence interval. The compound concentration required for the reduction of cell viability by 50% was determined by interpolation at 95% confidence. Comparisons between groups were performed with the student t-test and one-way analysis of variance (ANOVA) with the Bonferroni test on GraphPad Prism v.8.01 (GraphPad Software Inc., San Diego, CA, USA).

## 3. Results and Discussion

### 3.1. UV/vis Absorbance Spectrum

The interaction and bond creation between each compound involved in the formation of each delivery system type can change some absorption peaks along the UV/vis absorbance spectrum [29]. Taking this into account, it is possible to observe if the compounds are interacting with each other and how this interaction is changing the spectra. Thus, the UV/visible absorbance spectrum for each compound and each delivery system type is represented in Figure 1.

The UV/vis spectrum of pure quercetin shows only one significant peak near 374 nm in Figure 1A. This behavior has been reported in several other studies and it is regularly used to quantify the amount of quercetin that is present in the sample [20,30]. For SBE-β-CD/quercetin inclusion complexes, it is observed that the pure SBE-β-CD does not present any significant peak and no significant absorbance along the spectrum (Figure 1A). The sample resulting from the assembling of the delivery system evidences an increased absorbance along the spectrum (starting near 0.2), indicating that an interaction between the compounds has occurred. We can also observe in SBE-β-CD/quercetin inclusion complexes a peak similar to the one from free quercetin with decreased absorbance, supporting the incorporation of quercetin in the inclusion complexes.

As for SBE-β-CD/quercetin inclusion complexes coated with three different types of chitosan, represented in Figure 1A–C, we can observe significant differences between these systems and the inclusion complexes not coated in SBE-β-CD/quercetin. No significant peaks were observed in spectra of pure compounds, namely the SBE-β-CD and each type of chitosan, with the only significant peak present in the pure quercetin, as mentioned before. However, after the conjugation and formation of chitosan/SBE-β-CD/quercetin delivery systems, an increase in absorbance along all spectra was evidenced, indicating an interaction between the compounds of the systems [20]. This increase in absorbance was more evident for 5 kDa chitosan delivery system, suggesting that in this case there are present more small chitosan molecules to interact with the cyclodextrin complexes, increasing the general turbidity of the sample. The absence of a quercetin peak in these spectra suggests that the coating of the inclusion complex with the chitosan can protect the quercetin inside the inclusion complex cavity, avoiding its detection [20,31,32].

### 3.2. Characterization of SBE-β-CD/Quercetin Inclusion Complexes

After evidence of the formulation of the quercetin loaded systems, their properties such as the size, polydispersity index and surface charge (zeta potential) were determined at different concentrations of SBE-β-CD, as summarized in Table 1. Additionally, we also intended to reveal the best ratio of each used compound. The surface charge was only measured for ratios that presented a valid PdI.

The results presented in Table 1 showed that SBE-β-CD/quercetin inclusion complexes have large sizes for every tested ratio, probably due to the presence of too many negative charges. These charges came from SBE-β-CD. The SBE-β-CD-based system at a concentration of 3 mg/mL presented a size of 2468.33 ± 207.4 nm, a PdI of 0.123 ± 0.019 and a zeta potential of −21.03 ± 0.723 mV. The other tested ratios agglomerated instantly upon formation (effect visually observed) or presented a PdI value of 1, indicating that the sample has a wide range of sizes. Moreover, in these latter systems, the surface charge was not measured due to the high PdI of samples, which indicates a large range of nanoparticles sizes and therefore various surface charge peaks.

Therefore, it comes that the use of only SBE-β-CD leads to the formation of non-suitable systems for quercetin delivery to cancer cells.

Regarding the encapsulation efficiency of SBE-β-CD/quercetin inclusion complexes, and due to a phase-solubility study previously performed by H. Nguyen and F. Goycoolea, a maximum encapsulation efficiency of approximately 99.9% was obtained [20]. These results indicated that almost all quercetin was encapsulated inside the inclusion complexes and therefore no free quercetin was detected.

### 3.3. Characterization of Chitosan/SBE-β-CD/Quercetin Delivery Systems

Considering that SBE-β-CD/quercetin inclusion complexes showed non-suitable properties for in vitro studies due to the micrometric sizes and negative surface charges, coating with chitosan polymers was explored to increase the stability and surface charge and to enhance quercetin bioavailability. Chitosan is a natural polysaccharide widely used in the formulation of delivery systems due to its biocompatibility, biodegradability, low toxicity and high mucoadhesive ability [23,24,25]. In addition, the cationic nature and use of an acidic chitosan solution allows the interaction with the anionic SBE-β-CD, with the formation of nanosystems with enhanced properties that favor cellular internalization [20,21,33].

Different results have been reported according to characteristics of the used chitosan [26]. Chitosan can differ in MW, acetylation degree and chosen formulation method. Regarding their MW, the use of chitosan with a HMW, usually higher than 200 kDa, promotes the creation of systems with larger sizes, thereby allowing the encapsulation of bigger drugs. On the other hand, when considering the LMW chitosan, normally smaller than 150 kDa, the formed delivery systems exhibit smaller sizes. In this way, LMW chitosan has been widely used for the encapsulation of DNA vectors and small molecules, as well as for coating some delivery systems [20,26].

Taking into account this knowledge, in the present study, we have explored three types of chitosan, HMW (with a MW weight range between 200 and 500 kDa), LMW (with a MW range between 50 and 190 kDa) and 5 kDa. The use of different types of chitosan allows us to understand which one creates the most adequate delivery system, i.e., the one leading to increased quercetin loading capacity, carriage, solubility and, consequently, bioavailability in cervical cancer cells.

Chitosan/SBE-β-CD/quercetin delivery systems were optimized by changing the concentration of chitosan included in the formulations, namely in the range of 2–0.5 mg/mL. SBE-β-CD concentration was maintained constant at 3 mg/mL, considering the results described above. In total, seven different concentrations of each chitosan type (2, 1.5, 1, 0.9, 0.75, 0.6 and 0.5 mg/mL) were tested, as the respective outputs summarized in Table 2. Surface charge evaluation was performed for all the delivery systems conceived at ratios that give rise to a size lower than 500 nm, as this value was considered the biggest admissible [5,34].

Concerning the results presented in Table 2 for HMW chitosan delivery systems, it was observed that the ones formed from all chitosan concentration ratios below 1 mg/mL presented a nanometric size, an acceptable PdI (PdI lower than 0.4) and a zeta potential ranging from +38 mV to +42.7 mV. For the systems prepared with chitosan concentrations higher than 1 mg/mL, the obtained sizes are higher than 1000 nm, indicating an excess of positive charges and agglomeration, thus these formulations are not suitable for cellular application. The best results were obtained for delivery systems formed with concentrations of 1 mg/mL HMW chitosan and 3 mg/mL SBE-β-CD. Under these conditions, we obtained systems exhibiting a size of 272.07 ± 2.87 nm, PdI of 0.287 ± 0.011 and zeta potential of +38.0 ± 1.34 mV.

For LMW chitosan/SBE-β-CD/quercetin delivery systems, the same seven ratios were studied, but only four of them were completely characterized due to evident aggregation in chitosan concentration ratios of 2, 1.5 and 0.5 mg/mL. Surface charge studies were performed for all ratios that presented a size lower than 500 nm, following the same procedure as previously mentioned, being the results summarized in Table 2. As we can observe, the most adequate properties were obtained for the systems formulated with the concentration of 0.75 mg/mL LMW chitosan and 3 mg/mL SBE-β-CD, presenting a size of 269.0 ± 17.63 nm, a PdI value of 0.152 ± 0.092 and a surface charge of +42.3 ± 1.35 mV. These physico-chemical characteristics, namely a size smaller than 300 nm and a PdI value lower than 0.4, can enhance the cellular internalization capacity [34]. Along with this, we can consider these systems the most promising for subsequent in vitro studies. Moreover, concerning the obtained zeta potential, the nanosystems showed values significantly higher than +30 mV, being indicative of agglomeration risk [35,36].

The 5 kDa MW chitosan was also tested using the same concentrations and formulation method previously described, as evident visual agglomeration for the formulations made considering chitosan ratios of 2, 1.5, 0.6 and 0.5 mg/mL was observed. The other three ratios (1, 0.9 and 0.75 mg/mL) were characterized by assessing the size and PdI. The zeta potential was only measured for the ratio that has shown a size lower than 500 nm. The set of properties are presented in Table 2. Formulations containing 0.9 and 0.75 mg/mL of 5 kDa chitosan concentration exhibit an excessive size not suitable for cellular application, as previously mentioned. For the formulations made with the ratio of 1 mg/mL 5 kDa chitosan and 3 mg/mL SBE-β-CD the smallest size (282.6 ± 10.76 nm) and PdI (0.056 ± 0.030) were obtained. The surface charge indicated a value of +39.5 ± 2.01 mV.

According to the literature, several factors can influence the cellular internalization of delivery systems, the size, PdI and surface charge being the most relevant [26,34]. Several studies have reported that sizes between 100 and 200 nm are ideal for delivery systems. This size range is considered small enough to facilitate the cellular uptake, and avoid the clearance of too small sizes, which reduces the amount of drug that arrives at the target cells [34,37]. However, depending on the system constitution, the ideal size can be slightly different, particularly in the case of inclusion complexes that reported high delivery efficiency for nanosystems with sizes between 250 and 500 nm [5,34]. This fact is justified by the high complexity of inclusion complexes, being large enough to allow the encapsulation of large hydrophobic molecules inside it, such as quercetin. PdI is associated with the nanosystems size variation, and values lower than 0.4 are considered an indication of homogeneous samples [26].

In addition, another factor that can also influence target delivery and cellular internalization is the surface charge exhibited by the delivery systems [34]. It has been reported that positively charged nanoparticles are the most efficient for cell-membrane penetration and cellular internalization due to their effective binding to negatively charged groups on the cell surface [35,38]. Some comparative studies, using nanosystems with different surface charges have demonstrated that the charge significantly affects not only their internalization ability but also the cellular endocytosis mechanism, being the positively charged nanoparticles faster internalized [35,39]. Thus, in general, the formulation of delivery vectors with zeta potential values in the range between +10 and +30 mV are recommended, to favor their interaction with the cell membrane, without being excessively cationic to avoid toxicity and agglomeration [35,38,39]. Therefore, the results of size, PdI and zeta potential presented by the developed delivery systems are in accordance with other works previously reported on delivery systems formed by inclusion complexes conjugated with polymers [18,20].

Encapsulation efficiency studies were also performed for all the optimized delivery systems; the ones formed by considering the best ratios. The results indicated that quercetin encapsulation efficiency for each delivery system is approximately 99.9%.

### 3.4. Scanning Electron Microscopy

Another factor that can also influence cellular internalization is the morphology presented by nanosystems, with a regular and spherical morphology being desirable [40,41]. In this way, several images were captured by SEM for each type of formulated system and are presented in Figure 2.

Data shown in Figure 2 indicated non-uniform and non-spherical nanosystems for SBE-β-CD/quercetin inclusion complexes. As for chitosan/SBE-β-CD/quercetin delivery systems, all formulations showed uniform and spherical morphologies, and were therefore favorable for cell internalization.

Beyond morphology, the nanosystem’s size was also observed by SEM results. For the SBE-β-CD/quercetin inclusion complex, a size of approximately 1200 nm was observed, and for chitosan/SBE-β-CD/quercetin delivery systems, a size of approximately 200 nm was also detected. A size reduction between what was determined using Zetasizer and what was observed by SEM was noted in all delivery system types. This size reduction can be attributed to the drying process that delivery systems suffer in the SEM sample preparation procedure. This behavior has already been reported in several works [42,43,44].

### 3.5. Fourier Transform Infrared Spectroscopy

To further ensure that all compounds are present in each type of delivery system and to confirm if they are conjugated with each other, FTIR analysis of each system was also performed. In this way, the presence of specific chemical groups was evaluated for each compound, as well as the interaction between each compound in the respective system, and the spectra are shown in Figure 3.

Considering the FTIR results, pure quercetin presents a characteristic peak at a wavelength of 3386.22 cm^−1^, attributed to the phenolic stretching vibrations of the *-OH* bonds [45]. Additionally, several peaks in regions between 1600 cm^−1^ and 1100 cm^−1^ are also evidenced, corresponding to *-CO* stretching, aromatic stretching and bending and *-OH* phenolic bending [45].

As for the case of the pure chitosan spectrum, characteristic absorption peaks were identified at the wavelength between approximately 3272.42 and 3359.52 cm^−1^, corresponding to the intermolecular and intramolecular hydrogen bonds *-OH* and *-NH*; between 2873.78 and 2888.21 cm^−1^, corresponding to *-CH* stretching vibrations; and between 1027.41 and 1063.92 cm^−1^, attributed to the asymmetric stretching of the *C-O-C* bridge [26,29,46]. These wavelengths slightly vary according to the type of chitosan used.

Taking into consideration the spectra of pure SBE-β-CD, characteristic absorbance peaks were observed at the wavelength of 3387.42 cm^−1^, corresponding to the *-OH* bonds, and at the wavelengths of 1156.18 cm^−1^ and 1023.99 cm^−1^ corresponding to the *-CH* bonds and the asymmetric stretching of the *C-O-C* bridge [20]. After the addition of chitosan to the SBE-β-CD complexes, characteristic peaks of each compound were obtained, with only slight changes in the wavelengths of each peak arising from the interactions formed between the compounds.

In the case of chitosan/SBE-β-CD/quercetin delivery systems, a spectrum similar to the result without quercetin was verified, justified by the nature of the encapsulation system that is being explored. Briefly, cyclodextrins are inclusion complexes that allow hydrophobic drugs to be “trapped” inside their hydrophobic cavity, making the drug completely coated and limiting its detection by techniques such as FTIR [20]. Accordingly, the FTIR spectrum of the chitosan/SBE-β-CD/quercetin system allowed us to prove that quercetin was encapsulated inside the systems and not bound to their surface. This fact is also evidenced in other similar studies [20,47,48].

Furthermore, significant differences between each type of chitosan used were not observed, only some slight differences in peak wavelength, whether in the case of pure chitosan or for already formed systems that can be attributed to differences in MW of the chitosan used.

### 3.6. Release Studies

In order to understand how much quercetin was released from the viable delivery systems, release studies were performed over 4 days. Taking into account the characterization studies previously presented, SBE-β-CD/quercetin inclusion complexes were discarded as a viable option for the delivery of quercetin to cancer cells due to its excessive size and negative surface charge, making the process of cellular internalization harder [34,35]. In this way, chitosan/SBE-β-CD/quercetin delivery systems have been chosen for the release studies since they were revealed to possess promising physico-chemical properties. Release studies were performed at different pH values to mimic some conditions encountered in the human body. Thus, these assays were performed at pH 7.4, representative of the normal bloodstream pH, and a more acidic pH of 5.8, representative of the pH found in a tumor environment [29]. Taking into consideration the ideal characteristics of the delivery systems, we expect distinct release profiles for the two pH values tested. In this way, we expected a low quercetin release for pH 7.4, indicating that this type of delivery system is stable during the passage in the bloodstream and quercetin is maintained within the delivery system. When the systems achieve the tumoral site, where an acidic pH level is presented, a significant quercetin release should be obtained, thereby targeting these type of delivery systems to cancer cells [29,49]. The respective results for each chitosan/SBE-β-CD/quercetin delivery system can be seen in Figure 4.

All delivery systems showed an initial burst release effect that can be related to the freedom of drug molecules near the outer surface of nanosystems [50]. This burst release effect was followed by a sustained release that can be associated to the diffusion or time-dependent degradation of the delivery systems.

At pH 7.4, some chitosan/SBE-β-CD/quercetin delivery systems showed significant differences, as can be seen in Figure 4A. For HMW chitosan, a release of approximately 10% was observed after 96 h. However, for systems based on LMW and 5 kDa chitosan, a release of approximately 40% was observed. This result suggests that the HMW chitosan can promote a more stable conjugation with the SBE-β-CD by electrostatic interactions than the other two chitosans, probably due to the presence of more contact points in each chain of this polymer [20]. As a consequence, these delivery systems exhibit a slow biodegradability and a high stability that results in a low quercetin release in the bloodstream, especially for HMW chitosan based-delivery systems.

At pH 5.8, and as shown in Figure 4B, a cumulative release after 96 h between 50 and 60% was observed in the case of HMW and LMW chitosan delivery systems, respectively. However, for the 5 kDa chitosan delivery systems a higher release was observed, at approximately 75%. This increased cumulative release that was observed by decreasing pH can be attributed to the degradation of the delivery systems and quercetin escape under acidic conditions. Thus, these type of delivery systems show a significant release for a pH level representative of the tumoral environment, and a high stability in neutral pH, indicating the suitability of these systems to deliver quercetin to the targeted site [29,49].

Overall, these results showed that for an HMW chitosan-based delivery system it was possible to obtain an highly stable delivery system at normal pH as well as a significant release at an acidic medium representative of the pH found in the tumoral site. As for LMW and 5 kDa chitosan-based delivery systems, they had revealed good performance in quercetin release assays in both pHs studied, especially when the 5 kDa chitosan was used, reaching 75% of release at pH 5.8 after 96 h of incubation. Therefore, we achieved three delivery system types with different release profiles, with the HMW chitosan-based delivery system being the one that presented the better stability at pH 7.4 and the 5 kDa chitosan-based delivery system being the one that presented a controlled release ideal for drug delivery in a tumor environment.

### 3.7. Cell Internlization Studies

Each chitosan/SBE-β-CD/quercetin nanosystem was used to evaluate its capability for cell uptake and internalization. The images for an incubation period of 2 h are presented in Figure 5 and the nuclei stained blue are with DAPI and the chitosans of delivery systems are stained green with FITC.

According to the results of Figure 5, it was observed that all systems allowed the cell uptake and internalization after 2 h of transfection. The HMW chitosan/SBE-β-CD/quercetin delivery system showed the lowest fluorescence between all tested chitosan-based delivery systems, while LMW-based systems showed the highest one. These differences can be related to the different staining efficiencies of used chitosan types. However, all systems have the capacity of cell internalization and this type of delivery system can be suitable for quercetin delivery into cancer cells.

### 3.8. Cell Viability Studies

Quercetin’s therapeutic effect was evaluated for HMW, LMW and 5 kDa chitosan-based delivery systems and free quercetin as a positive control. Thus, these systems were used in cell viability studies with HeLa cells (HPV positive cervical cancer cell line), during 48 h (results presented in Figure 6).

Cell viability studies presented in Figure 6 demonstrated that the increase in quercetin concentration leads to a higher cellular viability reduction in HeLa cells. In fact, after an incubation period of 48 h, a significant cell viability reduction to 36.24% was observed with 150 µM of free quercetin. For lower concentrations of free quercetin, a cellular viability reduction to 82.81% was observed. In the case of HMW-, LMW- and 5 kDa chitosan-based delivery systems, the cell viability decreased to 20.12%, 14.43% and 6.94%, respectively, when the highest concentration was used (150 µM of encapsulated quercetin). For the lowest concentration of 0.1 µM, a cellular viability reduction of 79.27%, 89.85% and 84.27% was achieved, for HMW- LMW- and 5 kDa chitosan-based delivery systems, respectively. These data show that the encapsulation of quercetin with this type of delivery system is significantly more effective than the application with free quercetin, thereby potentiating its utilization in cancer therapy.

Therefore, the half inhibitory concentration (IC_50_) was determined in order to compare each system with free quercetin. Dose-response curves and final IC_50_ values for each formulation are indicated in Figure 6. According to the obtained results, free quercetin presented an IC_50_ of 59.84 µM, which is similar to values obtained by other studies [29,51,52,53]. As for chitosan/SBE-β-CD/quercetin delivery systems, an IC_50_ of 43.55 µM, 66.68 µM and 52.24 µM for HMW, LMW and 5 kDa chitosan were observed, respectively. Thus, chitosan/SBE-β-CD/quercetin delivery systems are capable of reducing cellular viability by 50% using generally lower quercetin concentrations in comparison with the free drug. In particular, HMW and 5 kDa chitosan/SBE-β-CD/quercetin delivery systems revealed a higher cytotoxic effect for cervical cancer cells in comparison with free quercetin. LMW chitosan/SBE-β-CD/quercetin delivery systems have shown a higher IC_50_. However, it is important to understand that these delivery systems perform a controlled release during more than 96 h, indicating that it can promote a more durable cytotoxic effect on cervical cancer cells than free quercetin. The results suggested that the use of HMW chitosan showed the best stability in neutral pH and can enables an easy release of quercetin at the targeted site if it presents an acidic pH. Hence, HMW/SBE-β-CD/quercetin delivery systems can be suitable to increase the bioavailability and effect of quercetin in cancer cells. Chitosan based delivery systems conjugated with SBE-β-CD inclusion complexes represent an effective method of quercetin encapsulation and thereby promote its controlled release in cervical cancer cells.

Comparing the obtained IC_50_ results with the ones from other quercetin delivery systems, there is an improvement in the therapeutic effect compared with nanoemulsions, micelles, and most synthetic polymeric systems [54,55,56,57,58,59]. In addition, our results are similar to liposome-based delivery systems, as well as systems formed by chitosan and quinoline, indicating an IC_50_ between 40 and 60 μM [29,51,52,53].

## 4. Conclusions

Considering the high anticancer activity of quercetin already reported in several studies, and taking into account that some bioavailability and stability problems have been associated with this drug, new strategies have emerged to enhance its effect in cancer cells. In the present work, SBE-β-CD-based delivery systems have been studied, with SBE-β-CD/quercetin inclusion complexes as well as three types of chitosan/SBE-β-CD/quercetin delivery systems (HMW, LMW and 5 kDa) being evaluated.

HMW chitosan/SBE-β-CD/quercetin delivery systems, based on the ratio of 1 mg/mL HMW chitosan and 3 mg/mL SBE-β-CD/quercetin, showed the best physico-chemical characterization results (size of 272.07 ± 2.87 nm, PdI of 0.287 ± 0.011, a zeta potential of +38.0 ± 1.34 mV) and an encapsulation efficiency of approximately 99.9%. Furthermore, SEM images also demonstrated a uniform and spherical morphology, ideal for cellular internalization of delivery systems. Release studies of this formulation indicated a quercetin release of 9.6% and 57.53% at pH 7.4 and pH 5.8, respectively, thereby being suitable for targeted cancer delivery due to its high stability at normal pH and significant release at acidic pH, representative of a tumoral environment. As for the other chitosan-based delivery systems, a significant release for both pH levels tested was observed, with the 5 kDa chitosan-based delivery system being the one that showed the highest release results, which were 41.41 and 76.64% at pH 7.4 and pH 5.8, respectively. Thus, this delivery system type has shown an important solution due to its controlled release and low initial “burst release effect”.

Cellular viability assays demonstrated that 5 kDa chitosan/SBE-β-CD/quercetin systems led to the greatest reduction in viability among the different systems tested. Subsequently, these systems revealed a lower IC_50_ than free quercetin, 52.24 and 59.84 μM, respectively, indicating an improved reduction of the HPV-positive cell viability. However, HMW chitosan/SBE-β-CD/quercetin delivery systems presented the lowest IC_50_ (43.55 μM), which is explained by the low quercetin release at neutral pH levels.

In summary, this work gave the first insights about the potential effect of the developed chitosan/cyclodextrins delivery systems to enhance quercetin loading capacity, carriage, solubility, release and consequently bioavailability and cell toxicity in HPV-positive cervical cancer cells. In addition, these delivery systems can be easily adjusted to efficiently deliver other hydrophobic drugs, solving the main limitation of the therapeutic effect of these kind of molecules. In the future, in vivo studies can be considered in order to assess the effect of quercetin delivered by developed nanosystems on cervical cancer therapy.

## Figures and Tables

**Figure 1 pharmaceutics-15-00936-f001:**
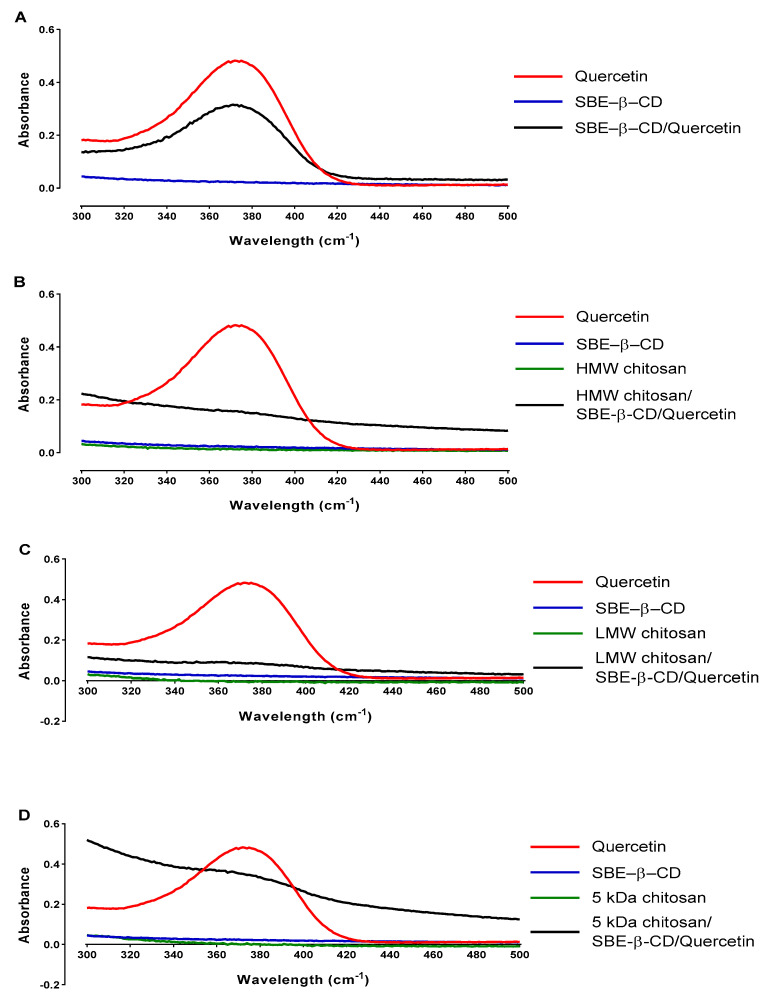
UV/visible absorbance spectrum of (**A**) SBE–β–CD/quercetin inclusion complexes; (**B**) HMW chitosan/SBE–β–CD/quercetin delivery systems; (**C**) LMW chitosan/SBE–β–CD/quercetin delivery systems and (**D**) 5 kDa chitosan/SBE–β–CD/quercetin delivery systems.

**Figure 2 pharmaceutics-15-00936-f002:**
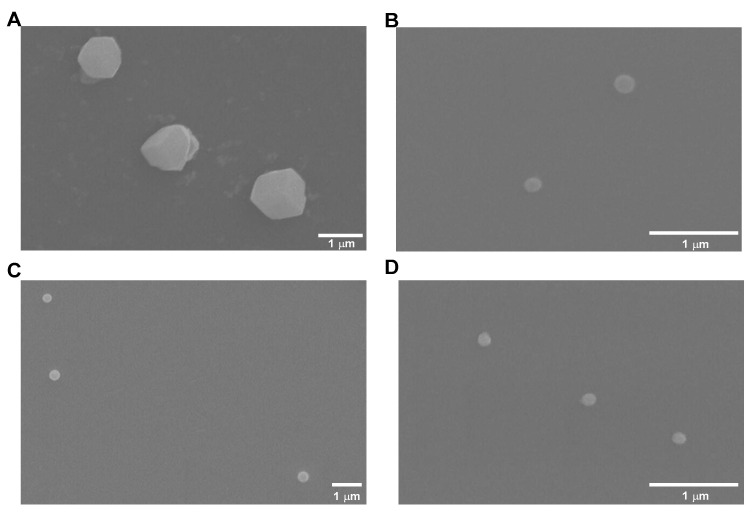
Selected images obtained by SEM for (**A**) SBE-β-CD/quercetin inclusion complexes; (**B**) HMW chitosan/SBE-β-CD/quercetin delivery systems; (**C**) LMW chitosan/SBE-β-CD/quercetin delivery systems; and (**D**) 5 kDa chitosan/SBE-β-CD/quercetin delivery systems.

**Figure 3 pharmaceutics-15-00936-f003:**
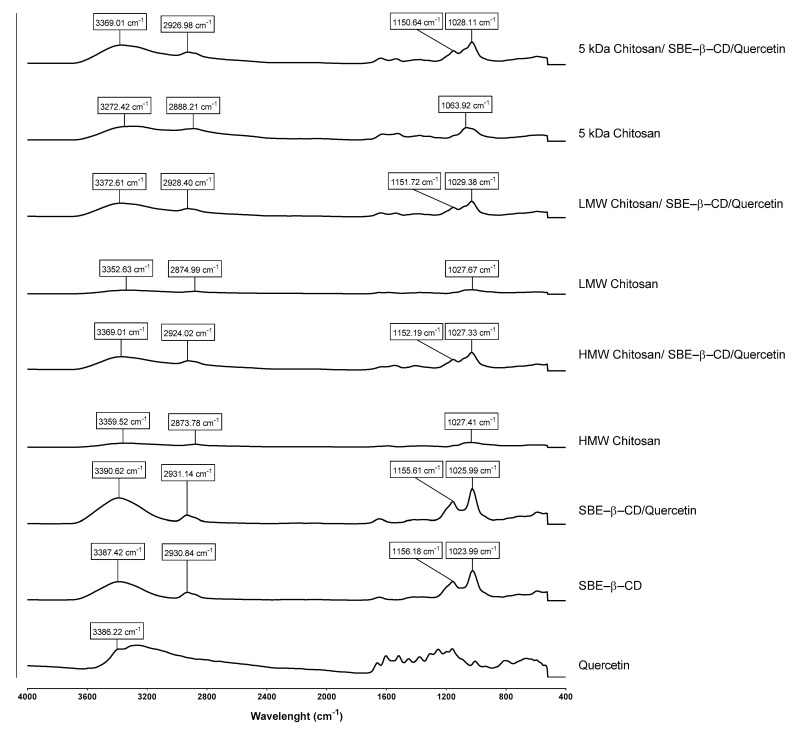
FTIR spectra (absorbance versus wavenumbers) of SBE–β–CD/quercetin inclusion complexes, HMW chitosan/SBE-β-CD/quercetin delivery systems, LMW chitosan/SBE–β–CD/quercetin delivery systems and 5 kDa chitosan/SBE–β–CD/quercetin delivery systems.

**Figure 4 pharmaceutics-15-00936-f004:**
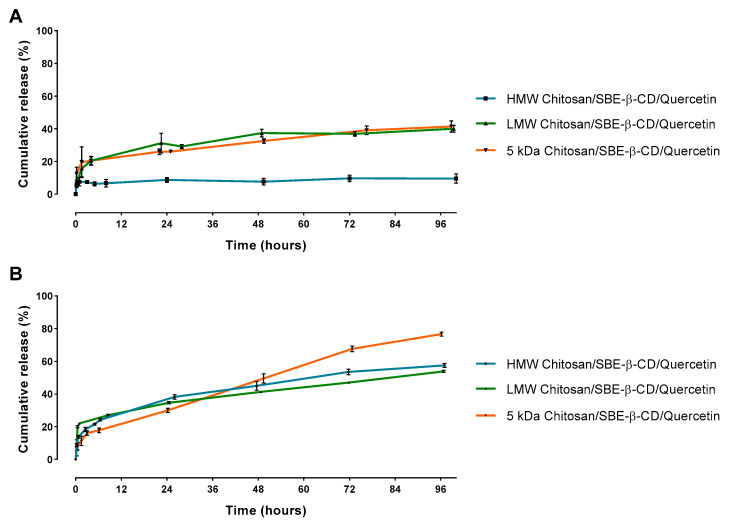
In vitro quercetin release studies during 96 h for each type of chitosan/SBE-β-CD/quercetin delivery systems at (**A**) pH 7.4 and (**B**) pH 5.8.

**Figure 5 pharmaceutics-15-00936-f005:**
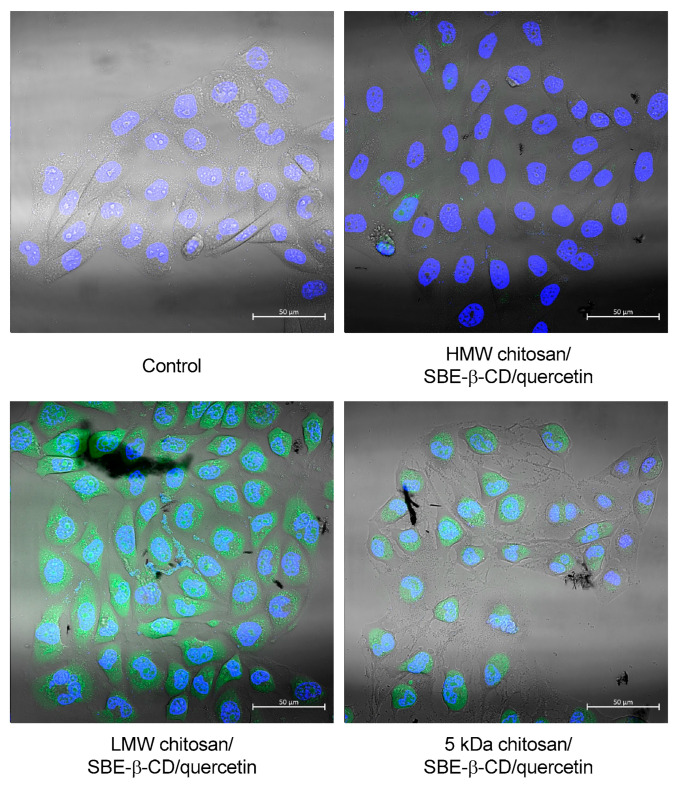
Uptake of FITC labeled chitosan-based delivery systems inside HeLa cells after 2 h of incubation. Each well was visualized under a confocal laser scanning microscope with fluorescence (green and blue) and bright field.

**Figure 6 pharmaceutics-15-00936-f006:**
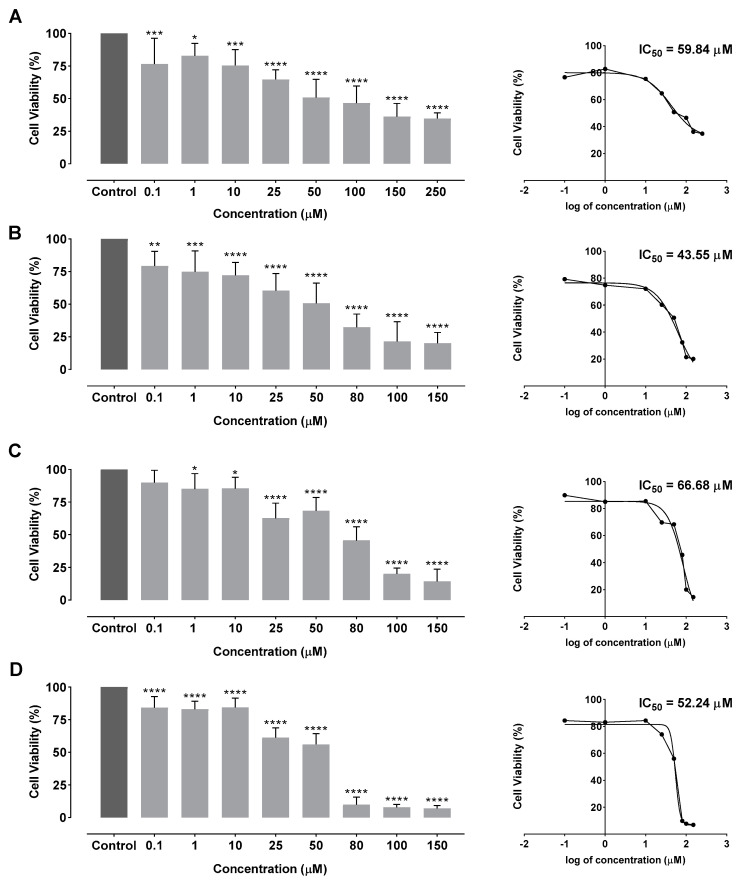
Cellular viability and dose-response curves on HeLa cells after 48 h incubation by MTT assay for (**A**) free quercetin; (**B**) HMW chitosan/SBE–β–CD/quercetin delivery systems; (**C**) LMW chitosan/SBE–β–CD/quercetin delivery systems; and (**D**) 5 kDa chitosan/SBE–β–CD/quercetin delivery systems. Non-transfected cells were used as control. Free quercetin and delivery systems were tested between the concentration range of 0.1 µM and 250 µM. Data are presented as mean ± SD for three independent experiments (n = 3) with four technical replicates and analyzed by one-way ANOVA with the Bonferroni test. Significance was determined as *p*-values * <0.05, ** <0.01, *** <0.001, **** <0.0001.

**Table 1 pharmaceutics-15-00936-t001:** Average size, PdI and zeta potential of SBE-β-CD/quercetin inclusion complexes for various ratios. The values are representative of the media obtained in three independent assays (n = 3). Abbreviations: ND = Not determined.

SBE-β-CD Concentration (mg/mL)	Size (nm)	PdI	Zeta Potential (mV)
6	2782.5 ± 974.6	1	ND
3	2468.33 ± 207.4	0.123 ± 0.019	−21.03 ± 0.723
1.5	1758 ± 410.6	1	ND

**Table 2 pharmaceutics-15-00936-t002:** Average size, PdI and zeta potential of HMW, LMW and 5 kDa chitosan/SBE-β-CD/quercetin delivery systems for various ratios. The values are representative of the media obtained in three independent assays (n = 3). Abbreviations: ND = Not determined.

Type of Chitosan	Chitosan Concentration (mg/mL)	Size (nm)	PdI	Zeta Potential (mV)
HMW	2	2714.67 ± 927.87	1	ND
	1.5	1379.5 ± 164.76	1	ND
	1	272.07 ± 2.87	0.287 ± 0.011	+38 ± 1.34
	0.9	299.6 ± 10.6	0.302 ± 0.032	+41.1 ± 1.85
	0.75	300.8 ± 4.291	0.249 ± 0.018	+42.7 ± 0.62
	0.6	296.7 ± 14.03	0.248 ± 0.014	+39.4 ± 3.99
	0.5	328.8 ± 19.37	0.218 ± 0.021	+38.1 ± 5.35
LMW	1	302.0 ± 17.80	0.285 ± 0.032	+47.4 ± 1.88
	0.9	275.9 ± 25.07	0.255 ± 0.044	+46.3 ± 1.35
	0.75	269.0 ± 17.63	0.152 ± 0.092	+42.3 ± 1.35
	0.6	1014 ± 776.8	0.732 ± 0.379	ND
5 kDa	1	282.6 ± 10.76	0.056 ± 0.030	+39.5 ± 2.01
	0.9	1002 ± 17.24	0.186 ± 0.022	ND
	0.75	18,920 ± 11,110	0.439 ± 0.487	ND

## Data Availability

Not applicable.

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
