# Peer review of "Development and Characterization of Quercetin-Loaded Delivery Systems for Increasing Its Bioavailability in Cervical Cancer Cells"

_pharmaceutics, 2023, doi:10.3390/pharmaceutics15030936_

Round 1

Reviewer 1 Report

1. Authors have done in vitro cell line study only, how they can say improving cervical cancer treatment without in vivo activity.

2. Section 2.2.1 must be elaborated. how the inclusion formed only by solubilization in water. how they have removed the water give reference. 

3. Give reference for section 2.2.2.

4. What is the significance of delivery system if it releases around 50% in 96h. How it will helpful in cervical cancer.

5. Rationale of the study not clear. 

6. Phase solubility data and graph missing.

Author Response

The authors would like to acknowledge the careful revision and pertinent reviewer’s comments. All the questions were answered in the attached document and the recommended modifications were made, being properly highlighted in yellow in the revised manuscript file. With this, we believe that we have further clarified the addressed comments and improve the manuscript.

Reviewer 2 Report

The authors describe their research in an important area of delivery of natural, biologically active quercetin molecule using multicomponent systems – nanoparticles composed of SBE-beta-CD and chitosan. Although, the increase in efficiency/toxicity of the systems developed is not drastically higher than that of quercetin itself, further future optimization might result in better delivery.

To the current manuscript I would have several comments:

Line 78: It is not clear what the authors mean by this sentence. The cyclodextrins' molecular diameters are much smaller, are the authors thinking of some aggregates or nanoparticles of CDs? Or is this specifically meant for sulfobutyl-ether-beta-CD?

The introduction part: If the authors consider delivery per os, is there any knowledge on transport mediation and effectivity by either of the components through intestinal membrane?

Figure 1: Why there is sudden increase of the baseline for 5 kDa chitosan/SBE-beta-CD/quercetin in the whole spectral range? This does not seem to originate from the quercetin itself or its interaction with other components. The remaining spectra do not provide much information except for showing the absorption of quercetin (Q) and the fact that Q is at very low concentrations in studied systems.

In the whole text: Please avoid using “superficial charge” and use more common term “surface charge”.

Table 1:  That’s a large standard deviation of size in the first experiment.

In general, the experiments where chitosan was added, shall there also be an information on the pH of solutions – this can be important as protonation of chitosan’s amino groups can have a large effect on interaction with SBE-beta-CD and overall formation and morphology of particles.

In general, have authors considered use of NMR spectroscopy for determination of particle composition?

Author Response

(The authors gave the same response as above.)

Reviewer 3 Report

The paper describes the development of chitosan/sulfonyl-ether-β-cyclodextrin(SBE-β-CD)-conjugated delivery systems loaded with quercetin as a potential alternative therapy of cervical cancer. The idea is interesting, although not exactly innovative, however the study seems very preliminary. In vitro efficacy could have been further investigated. The authors discussed a lot about cellular internalization however no studies regarding to that were performed. I recommend to perform a cellular uptake assay in order to investigate the efficacy of the delivery system.

Furthermore, some key revisions are required before the manuscript become suitable for publication:

-      Has a higher concentration of quercetin for encapsulation been tested? 0.3 mg/5ml seems to be very little thinking about further in vivo studies.

-        Why HMW chitosan/SBE-β-CD/quercetin delivery systems were discarded? Wouldn't a low release at pH 7.4 be interesting?

-        The text is a little confusing, especially topics 3.2 and 3.3, in which the results could be presented clearer.

-        Figure 1 – the lines are very similar, making it difficult to see the results.

-        Figure 4 - standard deviation is missing

-        How quercetin (free group) was solubilized in the cell viability assay?

-        NTA analysis could also be performed in order to investigate more about the diameter and size distribution of the delivery system.

-        In the text, authors wrote “PEG has well-proven cell toxicity limiting its use for in vivo assays”. Based on what references did the authors make this statement? PEG is widely used in vivo, being key component of several nanosystem.

-     According to figure 5, IC50 of the free drug is lower (59.84) than LMW chitosan/SBE-β-CD/quercetin delivery systems (66.68). Therefore the sentence “chitosan/SBE-β-  CD/quercetin delivery systems are capable of reducing cellular viability by 50 % using lower quercetin concentrations in comparison with the free drug” should be corrected in the text.

Author Response

(The authors gave the same response as above.)

Reviewer 4 Report

The authors describe the development and characterization of quercetin-loaded delivery systems to improve cervical cancer treatment. The physical-chemical characterization is just enough, but from a biological point of view, this study is not well done. For this reason, I suggest some experiment in order to improve the quality of manuscript. In this form, in my opinion, the manuscript is not suitable for publication in this journal.

How was loading efficiency studied? A calibration curve was done?

How is the mechanism of cellular internalization?

Confocal microscopy study could be useful to understand the intracellular localization.

How is the internalization efficiency? Confocal microscopy, spectroscopic study or cytofluorimetric assay could be useful to investigate this aspect.

From a biological point of view how this drug delivery system improves cervical cancer treatment? The values of IC50, estimated by MTT test are very similar. Justify this important point.

Author Response

(The authors gave the same response as above.)

Round 2

Reviewer 1 Report

Accept

Reviewer 3 Report

The authors have addressed all the comments properly, therefore I recommend to accept the paper. 

Reviewer 4 Report

The authors replied to all comments improving the quality of the manuscript. Now the manuscript is suitable for publication in this journal.